# Community Co-Design of Regional Actions for Children’s Nutritional Health Combining Indigenous Knowledge and Systems Thinking

**DOI:** 10.3390/ijerph19094936

**Published:** 2022-04-19

**Authors:** Pippa McKelvie-Sebileau, David Rees, David Tipene-Leach, Erica D’Souza, Boyd Swinburn, Sarah Gerritsen

**Affiliations:** 1School of Population Health, University of Auckland, Auckland 1023, New Zealand; boyd.swinburn@auckland.ac.nz (B.S.); s.gerritsen@auckland.ac.nz (S.G.); 2Research and Innovation Centre, Eastern Institute of Technology, Napier 4112, New Zealand; dtipene-leach@eit.ac.nz; 3Synergia Consulting Ltd., Auckland 1011, New Zealand; david.rees@synergia.co.nz; 4School of Future Environments, AUT University, Auckland 1141, New Zealand; e.dsouza@aut.ac.nz

**Keywords:** child nutrition, food systems, food environment, group model building, Indigenous knowledge, system dynamics

## Abstract

Children’s nutrition is highly influenced by community-level deprivation and socioeconomic inequalities and the health outcomes associated, such as childhood obesity, continue to widen. Systems Thinking using community-based system dynamics (CBSD) approaches can build community capacity, develop new knowledge and increase commitments to health improvement at the community level. We applied the formal structure and resources of a Group Model Building (GMB) approach, embedded within an Indigenous worldview to engage a high deprivation, high Indigenous population regional community in New Zealand to improve children’s nutrition. Three GMB workshops were held and the youth and adult participants created two systems map of the drivers and feedback loops of poor nutrition in the community. Māori Indigenous knowledge (mātauranga) and approaches (tikanga) were prioritized to ensure cultural safety of participants and to encourage identification of interventions that take into account social and cultural environmental factors. While the adult-constructed map focused more on the influence of societal factors such as cost of housing, financial literacy in communities, and social security, the youth-constructed map placed more emphasis on individual-environment factors such as the influence of marketing by the fast-food industry and mental wellbeing. Ten prioritized community-proposed interventions such as increasing cultural connections in schools, are presented with the feasibility and likely impact for change of each intervention rated by community leaders. The combination of community-based system dynamics methods of group model building and a mātauranga Māori worldview is a novel Indigenous systems approach that engages participants and highlights cultural and family issues in the systems maps, acknowledging the ongoing impact of historical colonization in our communities.

## 1. Introduction

Across the globe, socioeconomic inequalities are present for many non-communicable diseases, including obesity [1], and these inequalities continue to widen [2]. For children in Aotearoa, New Zealand, community deprivation is a major determinant of health outcomes and must be taken into account when tailoring interventions [3,4]. Indigenous Māori and Pasifika children are proportionately more likely to live in areas of low socio-economic advantage. The latest update of national nutritional indicators shows that community (dis)advantage and Māori or Pacific ethnicity are associated with a higher burden of obesity-related NCDs, food insecurity [5,6] and poor nutrition; with higher fast food and fizzy drink consumption and lower likelihood of eating breakfast 5 days a week [7]. These disparities in nutritional health outcomes are affected by the food environment of the community [8] including cultural, physical, economic and political factors [8]. 

Systems Thinking is a useful approach to understand and act on the prevention and improvement of NCDs such as obesity, taking into account health inequities [9]. It acknowledges the impact of the contextual environment and the complexity of interconnected environmental factors [10] and is well suited to address these complex problems characterized by non-linear cause and effect relationships interacting over time [11]. Systems Thinking, and the associated system dynamics, have been applied in public health research on chronic and complex issues such as childhood obesity prevention [12,13,14,15,16,17], particularly in high deprivation communities [18,19,20]. The benefits of these community-based system dynamics (CBSD) approaches can extend beyond the primary health outcome of interest and build local community capacity and desire for change [20,21], shift perspectives [17], increase collaboration, build community ownership, collective learning, and develop new insights [22]. 

Of particular interest for the present initiative, is a defining feature of complex systems, inter-connectedness, which is commonly recognized to overlap with ‘traditional wisdom’ [23]. CBSD methods have been applied with Indigenous communities to describe health innovations reflecting an Indigenous worldview [24] and recently, the methods were reported to be useful and appropriate for work in Australian Aboriginal communities, particularly with regard to self-determination [25]. 

The Hawke’s Bay region of New Zealand has a high Indigenous population where 37% of children are of Māori ethnicity, and there is relatively high community deprivation, with approximately 30% of the population living in the nationally determined quintile of highest deprivation [26]. Building on Gerritsen et al.’s [20,27] and Heke et al.’s [24] research, and the theoretical resonances between Systems Thinking and mātauranga Māori (Indigenous knowledge) [28,29], we hypothesized that the CBSD process using Group Model Building (GMB) would be an innovative and useful technique for the Hawke’s Bay community to collaboratively design actions to address children’s nutrition. GMB workshops train participants to think about the interconnections between core concepts, for example, ideas, people, objects and behaviors [24] and the emphasis is on collaboration and empowerment of participants through the process [30]. 

The present initiative, Nourishing Hawke’s Bay: He wairua tō te kai, aims to address the poor nutrition and health outcomes of children in the region, particularly the less advantaged Māori and Pasifika communities. It was designed around the broad principles of combining mātauranga Māori and Systems Thinking using CBSD. Initial engagement with stakeholders (especially Indigenous stakeholders) directed us to address food security in the community, to work with schools and to include Indigenous knowledge (mātauranga Māori) as a priority [31]. The primary aim of the series of GMB community workshops described here was to map the drivers of childhood malnutrition in the community and to collaboratively design community action to improve children’s nutrition. In addition, the process aimed to enhance coordination, build better understanding of the multiple and complex causes of poor nutrition in the region, build community capacity and initiate collective action by identifying areas in the system to intervene. This paper aims to describe the benefits of the approach we employed that combines Systems Thinking and Indigenous knowledge for community change. 

## 2. Materials and Methods

### 2.1. Study Design

This research is a qualitative case study reported following the consolidated criteria for reporting qualitative research (COREQ) checklist [32] as applicable and aligning with the CONSIDER statement for health research involving Indigenous peoples [33]. Ethical approval was obtained from the Eastern Institute of Technology Research and Ethics Approvals Committee, ref 20/03. Informed consent was obtained from all participants. 

### 2.2. Participants

Two groups of purposive sampled participants were invited to the GMB community workshops: stakeholders and rangatahi (youth) from local high schools. The New Zealand Ministry of Education’s decile system that classifies socio-economic advantage of school communities was used to select five low- to mid-decile (low to mid advantage) high schools to participate. School principals or senior teachers agreed to participate and selected youth in their final years (aged 16 years or older) who could attend the workshop series. Adult stakeholders were purposively selected based on the research team’s knowledge of individuals in the region with an interest in food security and resilience in high disadvantage communities through their professional roles or personal interests. Key stakeholders such as the District Health Board, Iwi (tribal group), school principals and Ministry of Education were represented and approximately half of all participants invited were of Māori ethnicity. Adult participants received an explanatory email and/or phone call and an invite to the three workshops. A koha (financial contribution) was offered to those who attended outside of their professional roles (e.g., volunteers from the food rescue organization).

In respect of individuals’ privacy and confidentiality, no demographic information was taken and individuals cannot be identified in these results.

### 2.3. Theoretical Framework to the Study

The methodology applied utilizes Community-Based System Dynamics [30] underpinned by system dynamics [34]. The GMB workshop processes developed for application in this community are based on the hypothesis that the holistic and relational view of Systems Thinking aligns with Indigenous knowledge and systems, in this context, mātauranga Māori [35] and they are grounded in the languages and practices of the community in which we were working. While we describe elements of this alignment throughout the present text, the theoretical resonances are described in McKelvie-Sebileau et al. [29]. 

### 2.4. Research Team Composition, Roles, Reflexivity and Cultural Safety

Members of the research team fulfilled six core roles in the workshops: the expert systems modeler (DR)—to lead the GMB activities with the group and to supervise and oversee the construction of the maps; the ‘pā whakawairua’ (DTL)—to hold the ‘mauri’ (essence) of the workshops and ensure that the occasion proceeds appropriately according to local Indigenous customs; the mātauranga Māori facilitator–to share Indigenous creation stories and knowledge as metaphors for the GMB tasks of the workshops; the note-takers (EDS, PMS, SG, BS) —to record the engagement and how the two processes of GMB and mātauranga Māori meshed together; and logistical support persons—to ensure the tables had all the stationery and other resources required for the tasks. All research team members acted as process facilitators and moved around tables to participate in the discussion and ensure participants had understood the tasks. All members of the research team had some experience working with Indigenous communities and were guided by the pā whakawairua, an experienced Māori public health researcher and the mātauranga Māori facilitator. The doctorate student member of the research team undertook training in kaupapa Māori research theory in parallel to the workshops to inform the theoretical positioning and ensure the methods we employed were kaupapa Māori-consistent. 

### 2.5. Group Model Building Process and Tools

The GMB community workshops followed a defined structure, both in terms of the series and the tools and tasks employed within each of the workshops. Firstly, a preparatory consultation workshop was hosted for selected stakeholders so that the research team could present the current state of knowledge about children’s nutrition and wellbeing in Hawke’s Bay, discuss the broad themes from the initial stakeholder consultation [31] and seek feedback on the proposed process of community GMB workshops. Nine stakeholders attended this informal preparatory engagement that was held via Zoom due to COVID-19 public health protections. 

Three structured community GMB workshops were then hosted in a large room at a community sports venue in July, August and September 2020 from 9 a.m. to 12 p.m. on a weekday. 

#### 2.5.1. Workshop 1: Defining the Problem

The goal of workshop 1 was to develop a shared understanding of the key issues impacting the nutrition of Hawke’s Bay children and youth and to introduce participants to the process of the GMB workshop series. Three scripts available from Scriptapedia [36] were utilized: (1) Hopes and Fears (as part of the ‘whakawhanaungatanga’‘getting to know each other’ process); (2) Reference Behavior Patterns/Graphs over Time (for participants to begin thinking about how key variables were increasing or decreasing dynamically over time); and (3) Causes and Consequences (to deepen their thinking about causal patterns). During the workshop tasks, participants were seated in groups of 4–6 around tables to work collaboratively on the activities. The youth and adult tables worked in separate groups on different sides of the same room. 

Tikanga (Māori custom) processes with respect to karakia (invocation customs), whakawhanaungatanga (getting to know each other), manaakitanga (welcoming and hosting, often through sharing food) and use of pūrākau (traditional Māori narratives) were prioritized in the workshop. Participants were invited to share a nutritious breakfast together and get to know each other or catch up with acquaintances in an informal setting before beginning the workshop. The pā whakawairua set the scene and ensured participants felt culturally safe. The karakia used to open and close the workshop were delivered in te reo Māori (language) then translated so that all participants, irrespective of language ability, could understand the intentions set. To assist participants to understand the process of constructing feedback loops, the mātauranga Māori facilitator invoked a Māori narrative (pūrakau) of the creation of clouds through evaporation resulting from interactions of Māori atua (deities) as a metaphor of systemic processes. 

#### 2.5.2. Workshop 2: Understanding the Problem

As in workshop 1, invocations/karakia were chosen to set the intention for the collaborative learnings of the workshop. In this second workshop, participants began co-construction of a visual map of the local food system through a series of ‘feedback loops’. Feedback loops are circular patterns of interconnection and influence between factors in a system. Activities were designed around understanding feedback loops and extending this thinking to look at how food environment structures are interconnected and influence each other, creating cycles that are difficult to break. Interventions positioned in feedback loops provide deeper understanding of the potential impact and pathways to achieve change within a system. The research team presented an integrated model of the linear Causes and Consequences developed in Workshop 1 and participants worked through the scripted ‘Causal loop diagrams’ activity [36]. Through this, they began to identify circular feedback loops between the constructs and to create causal loop diagrams. 

As an analogy with the GMB tasks participants were engaged in, the mātauranga Māori facilitator shared the Te Hekenga ‘great migration’ story where Kupe, the Polynesian navigator, led the flotilla of canoes from Hawaiiki (the original homelands) to Aotearoa (New Zealand). The desire for change and new opportunities in the homeland of Hawaiki alongside a need to map out the destination for better outcomes for their people were described as metaphors for the process of constructing and visualizing maps of the food environments in Hawke’s Bay. The navigation maps were required to journey to a chosen destination, which was presented as the ‘Hopes’ listed in workshop 1. The ‘Causes and consequences maps’ were described as current world maps-a description of the current state of children’s nutrition and to anticipate the perils, pushbacks and challenges on the journey; and the community-constructed Causal Loop Diagrams (CLD) were presented as the pathway to navigate. In addition, a whakatauki (Māori proverb): *Mā te kōrero, ka mōhio; mā te mōhio, ka mātau; mā te mātau, ka mārama* (from discussion comes awareness; from awareness comes knowing; from knowing comes enlightenment) was used as a discussion point emphasizing the need for community discussion and knowledge to produce community capacity building. 

#### 2.5.3. Construction of Systems Maps Using Community-Designed Feedback Loops

Following workshop 2, all concepts proposed by participants in the Causes and Consequences and CLD activities were entered into Vensim (SD modelling software, Vensim PLE x64, Ventana Systems, Harvard, MA, USA) to draft initial systems maps for each group (adults and youth). Firstly, each CLD produced by a table group was analyzed to identify the concepts with the most connections going in and out. Similar concepts (like household finances and household budget) were merged. Subsequently, each CLD was examined in relation to the other CLDs and they were integrated into one diagram using Vensim. As part of the refinement, some variables were discarded if they did not fit with any other CLD or were placed to the side for later consideration, and others were added if they were discussed during the workshop but not written into the CLDs produced by the table groups. Not all causal relationships (particularly from the linear Causes and Consequence activity) could be made into feedback loops but they were still represented on the maps. Using a system dynamics convention to explain causal relationships [37], CLDs were labelled as ‘reinforcing’ (indicating loops where change is compounded, often called ‘virtuous’ or ‘vicious’ cycles); or ‘balancing’ (where patterns are held in balance (good or bad) as one connection reduces the effect of another, much like a thermostat). Positive polarity (solid lines in the figure) indicate a positive relationship between the two variables (i.e., as one increases the other increases or as one decreases the other decreases), and a negative polarity (dotted lines) indicate an inverse relationship between the two variables (i.e., as one increases the other decreases or vice versa). Subsystems (themes) of related CLDs were then identified (indicated by colored lines on the maps). This iterative refinement process was undertaken by an experienced system dynamics modeler (DR) and the first author (PMS) to ensure the systems maps captured the community perceptions discussed in the workshop and the two final draft system maps were agreed by the full research team prior to presentation in workshop 3. 

#### 2.5.4. Workshop 3: Identifying Solutions

The primary aim of the third and final community workshop was to present the visual systems maps so that participants could make any adjustments or additions, before spending time to discuss and identify places in the systems to intervene and create a healthier food environment in Hawke’s Bay. Firstly, Indigenous narratives and knowledge were shared telling the whakapapa (origins) of kai (food) from a Te Ao Māori worldview, deepening all participants’ understanding of a holistic view of nourishment as well as an understanding of the factors like colonization that created the inequities present in the community today. This was important to cover prior to inviting participants to design the interventions. Participants described their interventions and indicated on the systems maps where and how the proposed action intervened in the system.

Following this, the systems maps were tidied in Vensim, simplified by amending variable names and editing the polarity of the connections to ensure the final systems maps conveyed participants’ intended messages. The two versions (adult stakeholders and youth) were checked by the full research team.

### 2.6. Post-Workshop Prioritization of Interventions

Combining similar interventions and removing out-of-scope interventions (non-community level), such as a national sugary drinks tax or reduced Goods and Services Tax on fruit and vegetable sales, the core research team presented a list of 10 final community-proposed interventions to the Nourishing Hawke’s Bay Steering Committee who scored each intervention for feasibility and impact.

### 2.7. Embedding Group Model Building Processes in Culturally Appropriate Methods

As described throughout this section, the GMB processes were combined with Indigenous processes and practices to embed System Dynamics within a Te Ao Māori worldview. As such, we developed an approach that was grounded in the language and practices of the community we were working with. Based on researcher observation and anecdotal reports from the participants, this “Indigenous Systems” approach was a natural fit and was a culturally safe way of engaging with the wide range of community participants. This approach is illustrated in Figure 1 with the structured and scripted GMB tasks that were inserted to fit within an Indigenous worldview and Indigenous practices described in the dark blue surrounding oval. 

## 3. Results

### 3.1. Participants

Over the three workshops, 19 rangatahi (youth) from five regional high schools, and 26 community stakeholders participated. The high schools comprised of two low decile (1–3) schools (low community advantage) and three mid-decile (4–7) schools (mid community advantage), with an average roll size of 576 (range: 293–956). Community stakeholders represented 24 organizations including the District Health Board, Ministry of Education, kaupapa Māori health providers and trusts, Iwi, Heart Foundation, Eastern Institute of Technology School of Health Science, Hawke’s Bay Community Fitness Centre Trust, Sport Hawke’s Bay, food rescue charity, local food production business representatives and a supermarket owner, as well as teachers from Early Learning Services and low advantage primary schools. We did not collect demographic information on the participants as this was not the objective of the study, however, we purposively selected Māori participants to ensure representation. Of the 26 adults participating, approximately half were of Māori ethnicity.

While the two groups (adults and youth) participated in the same activities, they were seated on different sides of the same room to allow them to explore their unique realities shaped by their specific contexts. Participants were invited to attend all 3 workshops but as these were held on weekdays, not all participants were able to engage in all three. 

### 3.2. Creation of the Systems Maps

Each table created between two and five CLD that were combined in the overall systems maps. See Appendix A for an example of a hand-drawn CLD that became a feedback loop in the adult systems map.

#### 3.2.1. Youth Food Systems Map and Subsystems

The youth visual map comprised of 13 reinforcing causal loops of factors influencing nutrition of children and youth in the region (indicated by R in Figure 2). These related to: social participation, physical activity, social hauora (wellbeing), cultural connections, intergenerational effects, income, money available, food production, cravings, marketing, time required for cooking and health. The CLDs were organized thematically into five major subsystems: the role of marketing and the power of fast-food industry on attractiveness of unhealthy options; money available, stress and general wellbeing (hauora); the power of intergenerational effects and modelling; cultural relationships and social participation; and the link between energy levels, physical activity and eating healthy food (Figure 2). 

#### 3.2.2. Adult/Stakeholder Food Systems Map and Subsytems

The map constructed by the adults comprised of 10 reinforcing CLDs (R in Figure 3) and 1 balancing CLD (B1 in Figure 3). These related to: access to healthy food, housing, financial literacy, budget, home cooking, mātauranga Māori (Indigenous knowledge), localized curriculum, price of fruit and vegetables and the need to work. The CLDs were organized around 5 major subsystems: ability to access healthy food; money available for food; ability and time available for home cooking; mātauranga Māori and possibility of initiatives to reduce cost of fruit and vegetables.

To describe the co-design process and ideas that had led to the proposed actions for intervention, we graphically simplified selected causal loops as a communication tool. These were used in particular for reporting back to and discussing interventions with the Steering Group. Figure 4 is an example of one of these using R9 from the adult map. 

### 3.3. Identifying Priority Interventions

Identifying places on the systems map for actions to intervene, the four youth groups proposed 29 potential interventions, with each group contributing between four and nine ideas. The four adult groups proposed 20 interventions with one group not listing any interventions on the map and the other three groups contributing between 4 and 8 interventions. After refinement by the research team removing out-of-scope interventions like a sugary drinks tax and combining similar suggestions, the full list of 49 interventions was reduced to 10 priority ideas expressed in Table 1 as potential areas for system re-orientation. 

These ideas were presented to the Steering Group consisting of school principals, senior management at the District Health Board and Ministry of Education, a kaupapa Māori trust representative, and the Heart Foundation. Each Steering Group member ranked the interventions for feasibility and impact. The average ranks were plotted on an impact matrix (Figure 5). Creating a network for schools participating in the new Ka Ora, Ka Ako healthy school lunch program to encourage best practice was considered to be both the most feasible and the most impactful (Table 1, action 1). The second most impactful was increasing Indigenous knowledge and cultural connectedness (mātauranga Māori) in schools, particularly around kai (food) (Table 1, action 4).

## 4. Discussion

The overall Nourishing Hawke’s Bay: He wairua tō te kai initiative was designed to engage the community in a shared vision for improved food environments in the region, eliciting community views on culturally-acceptable and equitable options for a systemic intervention in the local food system, as well as monitoring health and wellbeing of children in the region [38]. With the high Indigenous population in the community, the responsibilities of collaborative co-design, and with some previous experience in CBSD methods embedded within a mātauranga Māori worldview [24], we refined the model for use in this specific community and context. Not only is mātauranga Māori hypothesized to be a natural fit with CBSD, the approach aligned with the principles developed by community stakeholders during initial consultation that established the four initial pou (principles) for our work. These include: build in mātauranga Māori; consider hauora (holistic health); create opportunities for integration and cohesion; and work with the community; with food security and interventions in the school setting acknowledged as contextual factors [31].

The unique design of the GMB workshops with a group of youth and adults working on the same tasks allowed us to compare the two visual maps of the food systems “as described by youth” and “as described by adults”. Overall, the youth-constructed map placed more emphasis on individual-environment factors such as the influence of marketing by the fast-food industry that affected attractiveness and cravings for unhealthy food. Youth emphasized the importance of the feedback loop between emotional/mental health and consumption of fast-food. This is supported by Hemmingsson et al. who found emotional turmoil in childhood to be associated with subtle junk food self-medication to alleviate uncomfortable emotional states [39] and Hendriks et al. [18] reporting adolescents’ perceptions that psychological distress impacts on unhealthy food intake, body weight, and weight-related bullying. In parallel with our results, adolescents from five European countries using CLD to identify drivers of obesity [15] cited compulsive, addictive, binge- or comfort-eating, often as a way of coping with stress or other mental health issues. The CLD identified excessive body weight and/or poor perception of body weight that resulted in poor self-esteem and stress, all of which contributes to worsening mental health [15].

With regard to the impact of fast-food marketing on adolescents, it is unsurprising that this features in the youth CLD more strongly than in the adults’, given the heavy use of social media by the fast-food industry to target adolescents and the large amount of time they spend online [40,41,42]. Adolescents in Savona et al.’s work also identified the influence of large food manufacturers and advertisers as an strong driver of the consumption of unhealthy food [15]. Of particular concern is the concentration of advertising around commercial sites of sale [43], the concentration of fast-food outlets in low advantage communities [44], particularly schools [45,46], and the two-fold exposure of Māori children to unhealthy food and drink marketing compared to non-Māori children [47].

In comparison to the youth map, the adult map focused more on the influence of societal factors such as cost of housing, financial literacy in communities, and social security benefits which all influence health and the ability to access nutritious food. The recently introduced Ka Ora, Ka Ako healthy school lunch program [48] for low decile (low advantage) schools was seen as an important opportunity to break traditional patterns that have led to poor nutrition in the past. The individual wellbeing theme important for youth was not present in the systems map generated by our adult stakeholders. These drivers in the adult map, focusing mostly on cost of food, financial wellbeing and time available, align with other New Zealand work [27] that reported barriers to increased fruit and vegetable intake such as the high cost of fresh produce compared to fast food, parents having little time for food preparation, plus declining cooking skills and knowledge. A recent systematic review constructed CLD to synthesize existing evidence and identify system dynamics in the food system of low-income groups [49]. The five sub-systems of ‘geographical accessibility’, ‘household finances’, ‘household resources’, ‘individual influences’, and ‘social and cultural influences’ bear much similarity to the subsystems identified across our maps, particularly that of the adult stakeholders. 

Importantly, given the focus on Indigenous practices and knowledge in our methods, and perhaps as a result of this emphasis, both maps included the need for more cultural connectedness. Students highlighted the link between good cultural connectedness, social participation, and good mental health, while adults identified the need for increasing cultural knowledge at the school level and the potential for greater connection with culture to be an important lever for interventions, particularly in schools. The use of cultural connectedness in this research by embedding the CBSD methods within a Te Ao Māori worldview is a novel approach and this may have influenced participants to include cultural connection as a driver of positive health-promoting behaviors. In contrast to much obesity-prevention work that has been criticized as deficit based [50] the CLDs pertaining to cultural connectedness represent a strength-based approach, aligning with Indigenous values, culturally-sensitive and locally-specific [51]. 

The level of complexity illustrated in the maps created over three workshops is evidence that these methods engage participants to obtain a relatively deep level of understanding of the drivers of poor health in the community, encouraging them to think creatively and collectively about interventions. However, their complexity made it difficult to use with people who had not been involved in the workshops. As such, we developed the simplified visual representation of a CLD (Figure 4) which appears to capture the system dynamics notion of feedback and breaking reinforcing cycles, while being easily communicable and interpretable. While we did not empirically test the acceptability of the combined approach, we can only offer anecdotal indication of continued engagement with stakeholders, uptake of data collection in schools and community support for the initiative. With a mixed team of Māori and Pākehā (NZ European) researchers, this research does not follow a strict kaupapa Māori research design, but aligns with the priorities given to us by initial stakeholder engagement. Designed for and set within a specific community, the findings cannot be generalized beyond the time, place, and participants of the present study but they represent novel insights from youth and adult perspectives working collectively within a mātauranga Māori framework.

## 5. Conclusions

The poor nutritional intake of children in Hawke’s Bay observed in national monitoring initiatives is influenced by a number of drivers identified in the youth- and stakeholder-constructed systems maps. In Systems Thinking terms [23,30], poor dietary intake can be considered as “an emergent property of a complex adaptive system that sustains a food environment that increases the accessibility, availability, affordability and acceptability of unhealthy foods” [49]. Through a series of short workshops, participants were engaged to think deeply about such causes of poor nutrition and poor health, to map the drivers of these patterns and to work collectively to identify interventions. The two top prioritized interventions are currently being carried out by the Nourishing HB initiative working with community. The combination of community-based system dynamics methods of group model building and a mātauranga Māori worldview is a novel Indigenous systems approach that engages participants and highlights cultural and family issues in the systems maps, acknowledging the ongoing impact of historical colonization in our communities. 

## Figures and Tables

**Figure 1 ijerph-19-04936-f001:**
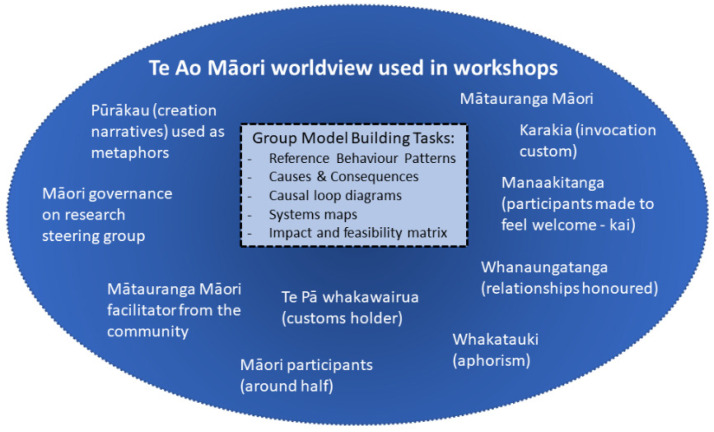
Group Model Building workshop tools and tasks embedded in a Te Ao Māori worldview (Mātauranga Māori–Indigenous Knowledge; kai–food).

**Figure 2 ijerph-19-04936-f002:**
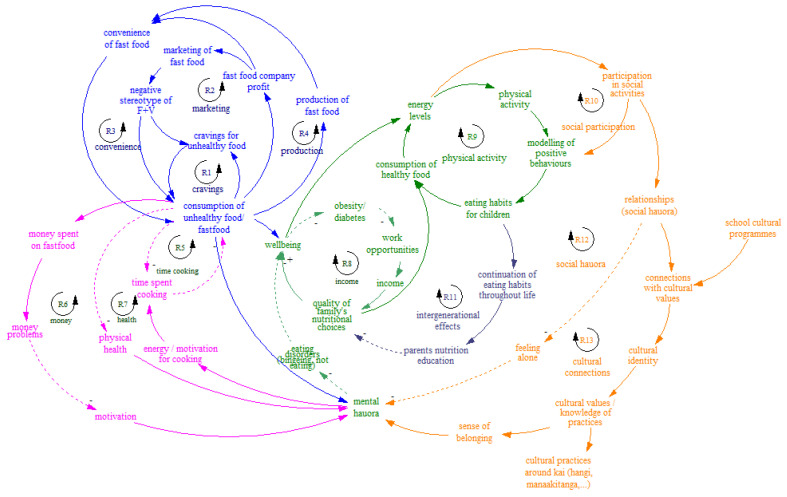
Youth–constructed food systems causal map.

**Figure 3 ijerph-19-04936-f003:**
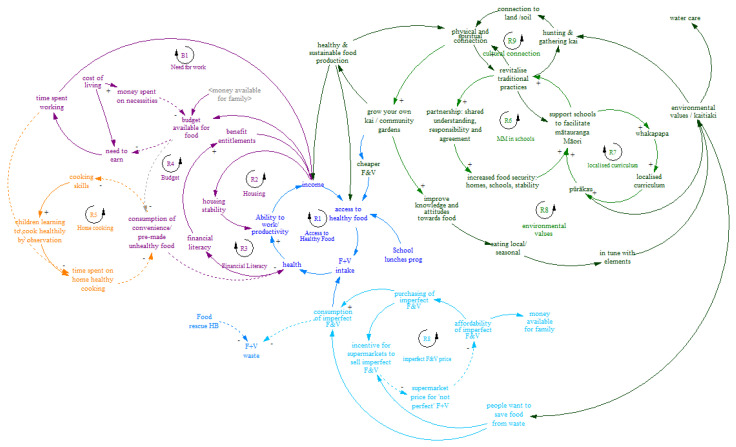
Adult–constructed food systems causal map.

**Figure 4 ijerph-19-04936-f004:**
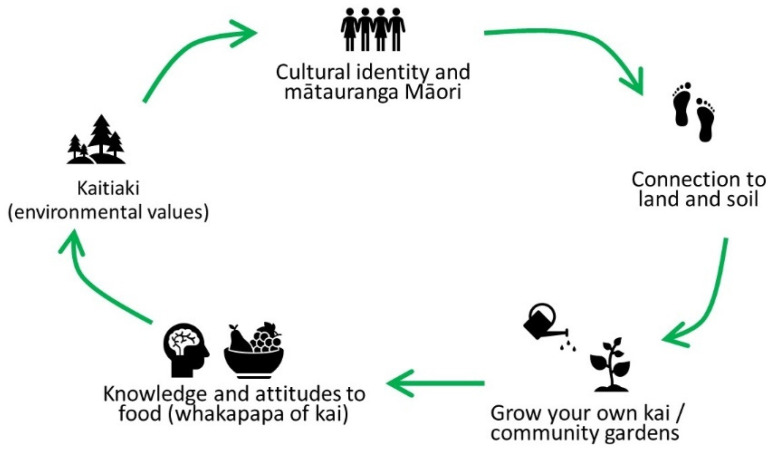
Simplified loop as a communication tool about the chosen intervention point (mātauranga = knowledge; kai = food; whakapapa = origins) (See Appendix A for the hand-drawn loop drafted by a participant during the workshops).

**Figure 5 ijerph-19-04936-f005:**
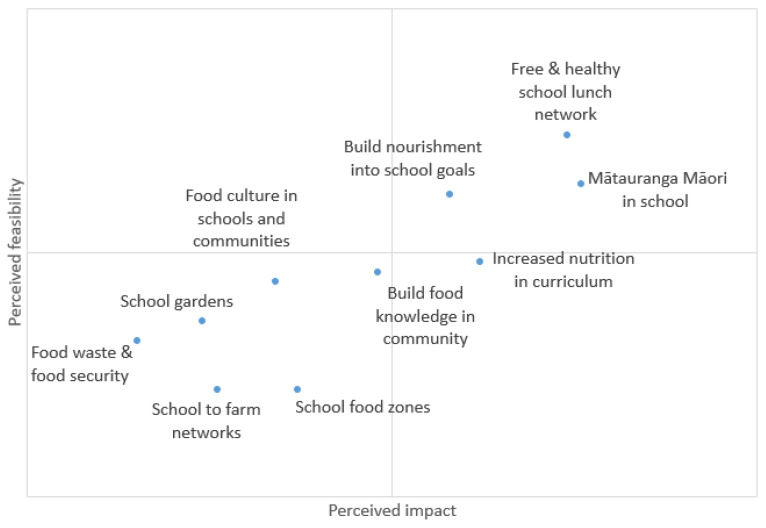
Impact matrix of interventions in the local food system (Mātauranga Māori–Indigenous knowledge).

**Table 1 ijerph-19-04936-t001:** Community-proposed interventions to promote resilient, health and food secure communities.

	System Reorientation		
Sub-System	From:	To:	Actions
1. Free & healthy school lunch network (Y-R9; A-R4; A-R4 *)	Loose network; competition model	Organized, sharing network; collaboration model, employment for the region	Build opt-in network of schools & local providers; ideas/stories sharing platforms (accessible website/database)
2. Food waste and food security (A-R3)	Food rescue to minimize waste; food banks to feed people in need; few interconnections; volunteer dependent	Food redistribution and use to minimize waste and food insecurity; distribution close to need.	Integrate food security, health and sustainability purposes into food rescue systems. Redistribution to schools.
3. Build nourishment into school goals (A-R7; Y-R9)	Feeding students with variable quality food—to fulfil service provision	Nourishing students with healthy, sustainable food to build resilient, young adults	Principals/School Boards to reflect nourishment in school goal/vision and policies to facilitate shared vision with school community
4. Mātauranga Māori ^1^ (MM) in school (A-R6)	Variable application; low priority	MM-inspired learning and doing throughout school	Tikanga Māori ^1^ integrated into processes/events; MM included in learning
5. Food culture in schools and communities (A-R9; Y-R13)	Variable priority; few cuisine experiences; lack of knowledge and understanding	Integrated priority; celebrated cuisine experiences; strengthen cultural identity and sense of belonging	Curriculum, events opportunities for experiencing cuisines from different cultures. Use this as an opportunity to educate and increase diversity in our diets.
6. School–farm networks (A-R8)	Few links to local farms, growers, orchards, gardens	Strong local networks including visits, learning, produce to schools, prizes	Events, link to school lunch network, curriculum inclusion, farmer/grower visits to schools
7. Increased nutrition in curriculum (Y-R11)	Some nutrition and food-related learning (mostly in lower grades)	Integrated mātauranga Māori, nutrition, food in curriculum, support from Ministry of Education	Identify opportunities in curriculum, teacher training sessions. Nutrition education across all grades. Internal rather than reliance on external providers
8. School food zones (SFZ) (Y-R2; Y-R3, A-R1)	Some in-school policies (overall weak and lack comprehensiveness), little engagement with food outlets around the school	500 m healthy school food zone includes in-school healthy food, no unhealthy food ads, agreements with local outlets in SFZ about serving kids junk food	Discussions with food outlets (e.g., Not serving students in uniform), pilot areas, feedback to parents
9. School gardens (A-R8; A-R9)	Some schools with gardens	Enhanced school gardens for learning, mātauranga Māori (revitalize traditional practices; whakapapa of kai ^1^), contribute to school lunches	Develop funding model for sustainability, gardener, māra kai, links with communities
10. Community food knowledge (A-R5, Y-R-13)	Some nutrition education in schools; informal learning about food	A rich variety of places in the community to learn about kai–heritage, growing, cooking, nutrition etc; mātauranga Māori	Look for multiple opportunities to add in food knowledge to events, curriculum, communications

^1^ Mātauranga Māori–Indigenous knowledge; tikanga Māori–customary practices; kai–food; whakapapa of kai–Māori perspectives on origins of food; māra kai–vegetable garden. * The letters in brackets indicate the CLD from the systems map when the actions intervene: e.g., A-R4 indicates that this action would affect Adult map reinforcing loop 4).

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
