# Peer review of "Community Co-Design of Regional Actions for Children’s Nutritional Health Combining Indigenous Knowledge and Systems Thinking"

_ijerph, 2022, doi:10.3390/ijerph19094936_

Round 1
Reviewer 1 Report
Interesting article on important topics related to consumer behavior in the food market. The authors' focus on a very specific market is both an advantage and a disadvantage. On the one hand, the relevant topic and detailed analysis, on the other, the risk of lower interest and application of the results among potential readers. Most of the authors of the publication describe the research methodology. Very difficult to analyze and develop conclusions. I suggest that the publication should describe the research methodology (on the example of a given sample). Although the authors present the results of the research in the text, table and diagrams, the lack of a more in-depth analysis is visible. There is a visible imbalance between the description and presentation of the methodology and inference. The research, especially the method of presenting the methodology, interesting schemas of results (a little not visible, the readability can be improved) is a great potential ... there was not enough room for better conclusions and presentation of relationships and dependencies.Author Response
Please see the attachment

Reviewer 2 Report
This study describes application of the community‑based system dynamics to children nutrition in a community in New Zealand. This is an important and timely topic, and I think this study can be valuable but it needs a major revision to clarify objectives of the study and its products.
Major comments:
- It is not clear what is the main contribution of this paper.
- Does it aim to map drivers of children malnutrition in the community? If yes, the maps produced by youth and adults need to be explained. Authors show the maps but did not describe the mechanisms identified in their study. I was very excited and wanted to know what mechanisms were identified and tried to read the maps myself but some words are not in English and could not follow the loops.
- Does this study aim to combine the community‑based system dynamics and mātauranga Māori worldview to come up with a new method to improve health in such communities? If yes, Figure 1 needs to be enriched. I am not sure what Figure 1 is trying to communicate at its current format. How the activities of a group model building (GMB) session in the light blue rectangle are related to those (non-English) words in the big dark blue rectangle?
- Does this study aim to show the differences in the maps generated by young and adult participants?
- Does it aim to identify interventions that are perceived to be effective? If yes, it would be interesting to show where on the map, these interventions affect the system (connect interventions and the maps). In other words, use system dynamics approach and the maps, to show why these interventions can systematically improve children’s nutrition (i.e., indicate how interventions affect the causal loop diagram).
The authors might have many of these items in mind but they should be clearly mentioned and the results that they present should represent the objectives
- I think one of the strength of this study is adjusting the community‑based system dynamics to the culture of the community. This is very valuable. I think the paper should highlight it but does not use many non-English words (local language) in this paper for international audience. It has reduced the readability of this paper and it is difficult to follow it. For example, the authors used “youth” or “student” in English and rangatahi to refer to their young participants. Use consistent terms and avoid using so many non-English words to improve readability.
Minor comments:
Page 1, line 18: Group model building is an approach. I am not sure what you mean by “we applied formal structure and resources of group model building tasks”. You can simply say you applied group model building approach to engage participants and map drivers of children malnutrition.
Page 1. Line 24: strengths-based interventions: As you mentioned in Figure 5, “perceived impact” was used to prioritize intervention. I am not sure if it can be claimed that strength-based interventions were identified.
Page 2, line 94: Spell out COREQ
Page 3, line 103: change “was use” to “was used”
Page 3, line 119: Have you developed methods? It seems to me that this study applies GMB to a case. Change your sentence or clarify what method was developed.
Page 4, line 157: What do you mean by “see below”
Page 3, section 2.2: You may want to mentioned number of participants in the method, not result.
Page 5, line 188-190: This sentence is not clear. Please restructure or use punctuation to make it readable.
Page 5, line 221: indicate which figure
Page 5, line 228: change “were agree”
Round 2
Reviewer 2 Report
The authors did not significantly improve the paper.